# Enabled Artificial Intelligence (AI) to Develop Sehhaty Wa Daghty App of Self-Management for Saudi Patients with Hypertension: A Qualitative Study

Adel Alzahrani [1],* , Valerie Gay [1] and Ryan Alturki [2]

1 Faculty of Engineering and Information Technology, University of Technology Sydney, Sydney, NSW 2007, Australia; valerie.gay@uts.edu.au
2 Department of Information Science, College of Computer and Information Systems, Umm al-Qura University, Makkah 24382, Saudi Arabia; rmturki@uqu.edu.sa
* Correspondence: adel.s.alzahrani@student.uts.edu.au

**Abstract:** (1) Background: The prevalence of uncontrolled hypertension is rising all across the world, making it a concern for public health. The usage of mobile health applications has resulted in a number of positive outcomes for the management and control of hypertension. (2) Objective: The study's primary goal is to explain the steps to create a hypertension application (app) that considers cultural and social standards in Saudi Arabia, motivational features, and the needs of male and female Saudi citizens. (3) Methods: This study reports the emerged features and content needed to be adapted or developed in health apps for hypertension patients during an interactive qualitative analysis focus group activity with ($n$ = 5) experts from the Saudi Ministry of Health. A gap analysis was conducted to develop an app based on a deep understanding of user needs with a patient-centred approach. (4) Results: Based on the participant's reviews in this study, the app was easy to use and can help Saudi patients to control their hypertension, the design was interactive, motivational features are user-friendly, and there is a need to consider other platforms such as Android and Blackberry in a future version. (5) Conclusions: Mobile health apps can help Saudis change their unhealthy lifestyles. Target users, usability, motivational features, and social and cultural standards must be considered to meet the app's aim.

**Keywords:** blood pressure; hypertension; mHealth; self-monitor; Saudi Arabia





## 1. Introduction

Chronic diseases such as hypertension, also known as high blood pressure is, a silent killer. It is a global health issue that requires increased care and blood pressure monitoring [1]. It is estimated that over 1.4 billion individuals worldwide have hypertension, which corresponds to one-third of all adults [2]. Untreated or uncontrolled hypertension is the single most crucial factor in the development of cardiovascular disease, which includes conditions such as coronary artery disease, heart failure, and stroke [3]. Saudi Arabia is among the countries with a high prevalence of hypertension. It ranks fourth globally for the number of people affected by hypertension [4]. The prevalence of hypertension (HTN) among adults in Saudi Arabia has been reported to be 17% [5].

The management of hypertension remains poor, although hypertension is a significant contributor to cardiovascular disease [6]. A study revealed a lack of evidence of managed care in Saudi Arabia, including hypertension patient involvement in disease management, suggesting that steps are needed to empower patients to take a greater role in disease management [7]. Therefore, hypertension self-management can significantly improve blood pressure control in patients with hypertension [8,9]. Hypertension can be significantly lowered by making healthy lifestyle adjustments such as limiting sodium intake, losing weight, and increasing aerobic exercise [10]. Therefore, mobile health applications, also

known as mHealth, provide a way to monitor a patient's health conditions, including diet, body weight, physical activity, and blood pressure.

The speed and efficiency with which data may be processed and shared have improved due to rapid advancements in digital computing and mobile communications technology. In particular, the increased availability of smartphones has increased the development and use of smartphone applications (apps). In the past, mobile health interventions used voice or text-based short message services (SMS). However, the growing availability and ease of use of apps have led to a significant increase in mobile applications that can be used to change health behaviours [11]. This is supported by [12], a study that indicated the growing field of mHealth and its use of new technologies could help improve healthcare and encourage people to live healthier lives. Mobile health applications have resulted in various beneficial effects for patients [1]. Therefore, health-related mobile communication applications (mHealth) are gaining popularity throughout the world due to their ability to help with behaviour change by encouraging healthy practices on a regular basis [13]. In addition, the mHealth app is already making it easier for patients to obtain treatment and professional assistance [14].

AI is an abbreviation for "Artificial Intelligence", which describes a group of technologies that enable computers and machines to simulate human intelligence [15]. AI technology has advanced significantly, particularly in the areas of machine learning, data analysis, speech recognition, and robotics [16]. According to [17], building a mobile health app that is abundant in AI features is difficult for developers; however, it can use AI to help patients to manage their disease and reach their goals. Voice assistants and chatbots are two examples of AI virtual assistant apps that allow users to interact with the apps and obtain the information they need quickly and easily [18]. We included AI in the Sehhaty Wa Daghty app, which in English language means My Health and My Blood Pressure, with a voice assistant to help users navigate more efficiently throughout the app in screens.

This paper aims to explain the development of the Sehhaty Wa Daghty app of self-management for patients with hypertension in Saudi Arabia and how it can use AI to help patients to manage their diseases. The Sehhaty Wa Daghty app aids Saudi adults with hypertension in self-managing their condition and enhancing their health and fitness levels.

### 1.1. Previous Work

To ensure that the Sehhaty Wa Daghty app is suitable for patients with hypertension in Saudi Arabia, we conducted semi-structured interviews in the previous paper [4] that showed what people expect from health monitoring apps and how they use them. It indicated that users could help designers make health apps that encourage self-monitoring and persistence, which leads to the better self-management of hypertension, by telling them about their different experiences and what they hope to get out of possible future interventions. The mobile health apps should be able to track physical activity, give information about diet, and send reminders. This will help patients stick to healthy lifestyles as well as help hypertension patients take better care of themselves. Mobile health apps can help individuals with high blood pressure manage and track their health. The study revealed that mobile health apps should not share their users' information and should have better privacy warnings. Therefore, this study considers all user perspectives and the needs of Saudi individuals in developing a hypertension management mobile technology solution.

### 1.2. Objectives

This study aims to describe the development process of the software application called Sehhaty Wa Daghty app, the tools used to design the app, the challenges encountered by the designers, and the lessons learned throughout the development process.

## 2. Methods

In the first phase, during the development of the Sehhaty Wa Daghty app, a total of ($n$ = 21) participants were interviewed. The participants consisted of doctors and patients receiving treatments for hypertension at the Ministry of Health. In the second phase, a focus group of five expert participants were conducted to collect current and emerging provider-focused implementation tools and resources for developing an app that meets the needs and preferences of hypertension patients. Based on phase 1 findings, a gap analysis was conducted to develop an app based on a deep understanding of users' needs with a patient-centred approach, as well as to determine what features and content are needed to be adapted or developed.

The first author (AZ) facilitated the focus group conversation using an open-ended interview question to guide the discussion. The qualitative questions from the focus group were reported by looking for the Saudi people's viewpoints and needs to build a hypertension management mobile technology solution. All participants were Saudi, and they shared feedback and suggestions about the app. This study reports the emerged features and content that needed to be adapted or developed in health apps for hypertension patients during an interactive qualitative analysis focus group activity with five experts from the Saudi Ministry of Health. Informed-consent procedures were explained at the beginning of the meeting. The focus group discussion was recorded with the permission of the participants.

### 2.1. The Development

The Sehhaty Wa Daghty app is an Arabic health app with three main components: blood pressure, physical activity, and food consumption. This app focuses on designing and developing a mobile technology solution to self-monitor HTN and improve health and fitness levels among Saudi adults. It is mainly developed for iPhones.

During the development of the Sehhaty Wa Daghty app, the initial focus group consisted of five participants who contributed within the initial focus group. Their information is shown in Table 1. All of them are Saudi citizens and suffer from hypertension or deal with it. Nevertheless, they are very different from one another in terms of age, occupation, and level of education. According to [19], it is advisable to have five to ten diverse individuals in terms of age, occupation, and experience within a focus group. A recording device was used during the focus group. The approval to conduct this study was granted by the ethics committee of the University of Technology Sydney, UTS HREC REF NO. ETH21-6571.

**Table 1.** Participants in focus group information.

| Gender | Age | Education Level | Type of Smart Phone |
|---|---|---|---|
| Male | >60 | Undergraduate degree | iPhone |
| Female | 41–59 | Diploma | iPhone |
| Male | 18–29 | High School | iPhone |
| Female | 30–40 | Postgraduate degree | iPhone |
| Male | 41–59 | Postgraduate degree | iPhone |

When design elements or app features were developed, they were shared with the participants within the focus group via TestFlight. TestFlight is an application that enables iOS developers to invite users to beta-test an application that is currently being developed. Following the completion of the testing phase, a group conversation was held to collect the opinions and suggestions of the focus group members. The application's overall quality was significantly enhanced as a direct result of the implementation of this method.

### 2.2. What Is the Sehhaty Wa Daghty App?

It is an Arabic health app that tracks blood pressure, physical activity, and diet. The Sehhaty Wa Daghty app aims to monitor and control blood pressure and to keep track of a patient's lifestyle, such as physical activity and food consumption. It also provides medical advice in order to improve the patient's health as well as the process of controlling the patient's blood pressure.

The app provides data and reminders for treatment, walking, and drinking water, as well as alerts for prayer times and the nearest mosque to your location. There are some voice commands in the Artificial Intelligence phase of the application. This application concerns Saudi patients and mainly focuses on Saudi culture to improve their lifestyle and raise their awareness.

The app is designed only for iOS devices in the initial stage and is an easy-to-use and interactive app. It also considers the Saudi individuals' perspectives and needs to design a hypertension management mobile technology solution. It similarly includes medical experts in developing the Sehhaty Wa Daghty app to improve its quality. The app's name, logo, and slogan were developed with the help of an initial focus group from the Saudi community (Figure 1).

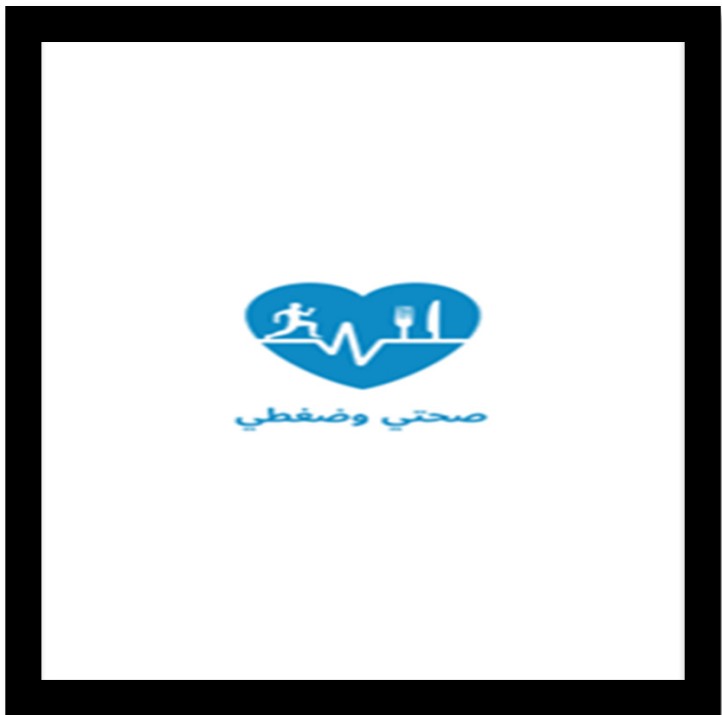

**Figure 1.** The logo and slogan of the Sehhaty Wa Daghty app (My Health and My Blood Pressure).

Moreover, there is a welcome message, suggested with the focus group, when you open the app, which says: "Your health is valuable: maintain it", as shown in Figure 2.

The membership and social networking features are available to users when they first start using the Sehhaty Wa Daghty app (Figure 3). If you already have an account, the app asks you to provide your mobile number, and then you will receive a code to verify and then log in to the app. If you do not have an account, the app has two options to log in to the app: first, by creating a new account by finishing a two-step registration process that involves creating a user profile and then entering user information data and body measurements, and, second, by choosing to sign in via a Twitter or Facebook (Apple) (Google) account and then completing a one-step registration process that involves entering user information data and body measurements.

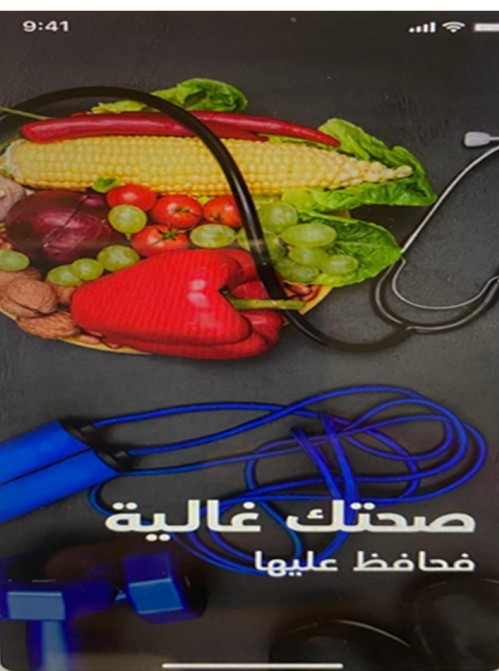

**Figure 2.** Welcome massage (Your health is valuable: maintain it).

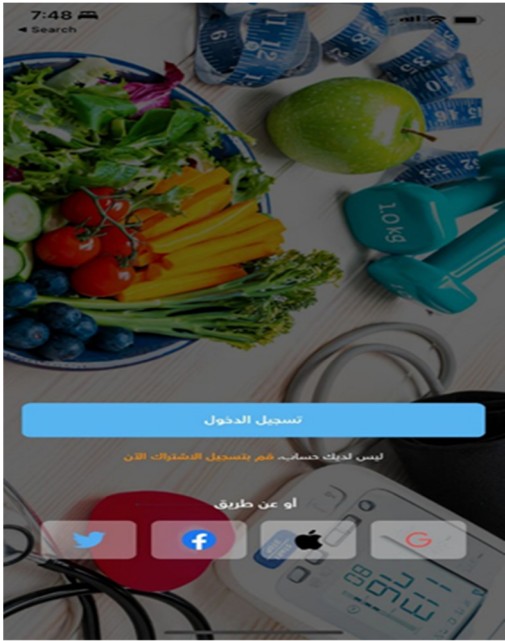

**Figure 3.** Membership and social networking features.

When creating a user profile, users are required to provide basic information such as their name and email address, and choose a password for their profile. Users are required to input their gender, age, height in centimetres, weight in kilos, and ideal weight if known.

After you log in to the Sehhaty Wa Daghty app, the user can see the main interface (Figure 4) for the app that contains a summary of your daily activity and five menu bars such as User Account, Chat, a plus sign, Advice, and Return to the Main Interface. The plus sign takes you to three main sections: Add Blood Pressure (BP) (Figure 5), Water Consumption, and Diet. In the main interface, the user can see the summary of his/her daily activity. The summary includes five charts: steps, calories, water consumption, blood pressure, and physical activity (Figure 4).

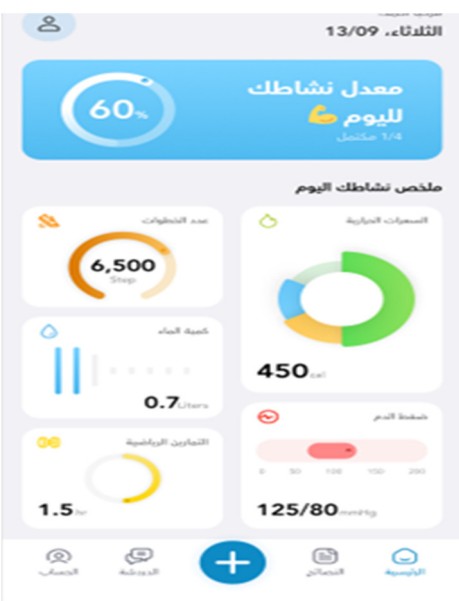

**Figure 4.** Main interface.

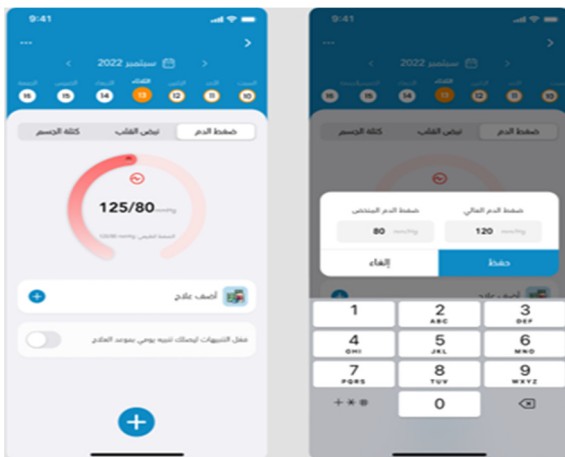

**Figure 5.** BP.

In the previous paper [1], we divided the themes into four main categories: Perceived Health Benefits of the App, Features and Usability, Suggestions for App Content, and Security and Privacy.

2.2.1. Perceived Health Benefits of the App

This first theme includes three subthemes: Physical Activities, Related Diet Information, and Reminders.

1.　Physical Activities

The Sehhaty Wa Daghty app's physical activity section was designed with exercises suitable for patients with hypertension in Saudi Arabia, and they can perform them at least at home. This app has motivational features encouraging users to perform physical activities such as Let's Walk, Train, and Pray.

These three unique features can encourage users to walk and exercise more as well as remind them to pray. The first one is Let's Walk. It encourages users to walk together as a group every week as well as monitor everyday walking. Users are able to choose and vote between three places to gather and start walking as a group. Therefore, you can increase your daily activity by keeping track of your walks and sharing them with users (Figure 6).

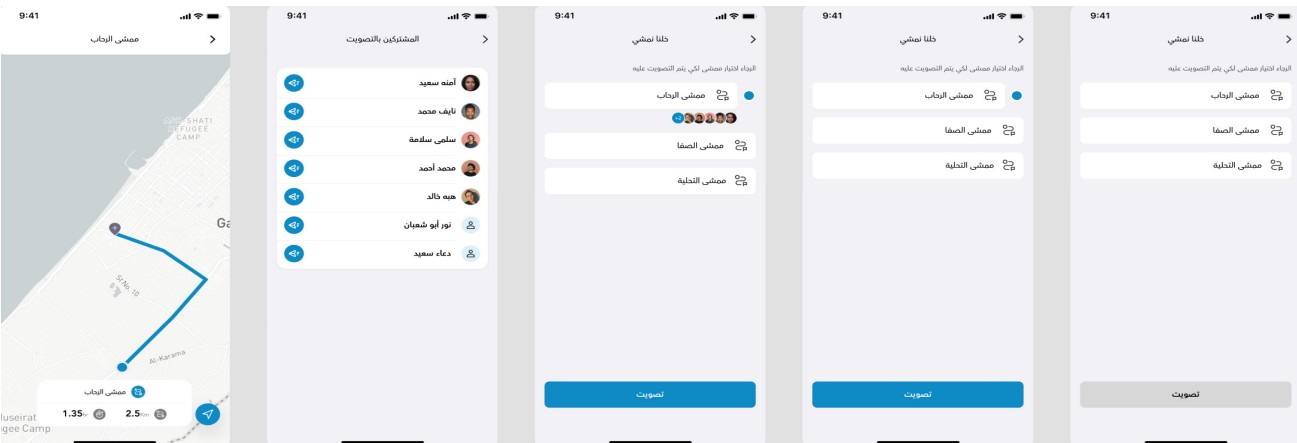

**Figure 6.** Let's Walk feature.

The second feature is Let's Train. Training should be the easiest thing in the world, but how often have you begun a new diet or exercise plan only to find that you have returned to your old ways after the first week? Losing weight can be challenging, especially if the users, such as the Saudi community, are busy with family and work commitments.

Thus, the best strategy for weight loss is to train. It is not necessary to spend countless hours in the gym in order to lose weight. You will burn more calories, sweat more, and lose more weight than ever by utilising tried-and-true methods designed to maximise each session's results. Therefore, the Let's Train feature includes special exercises designed for hypertension users that are recommended by medical experts in the Ministry of Health for Saudi patients. Therefore, it is easy to perform those exercises at home as Saudi Arabia has a hot climate during the year that prevents users from exercising outdoors.

The third feature is Let's Pray. More than 93% of the people in Saudi Arabia are Muslim [20]. Most Muslim men regularly visit mosques to fulfil their religious obligation of praying five times daily. As a result, this feature enables users to activate notifications or alerts when it is time for prayer, displaying the mosques closest to the user's current location (Figure 7). If the user selects this option, the app will direct them. The software will provide users with a selection of mosques, allowing them to go to a mosque further away if they wish to increase their walking distance.

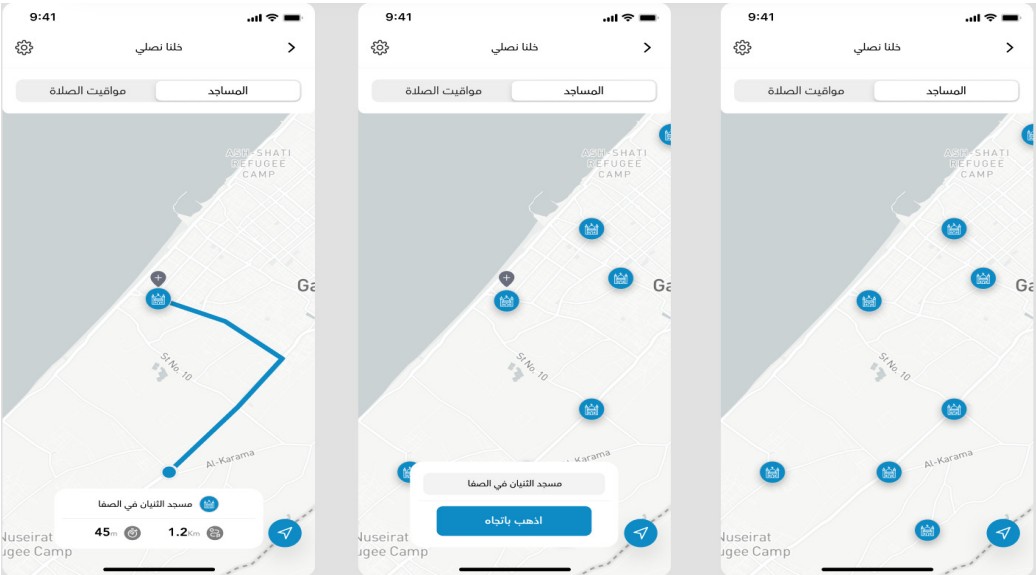

**Figure 7.** Let's Pray feature.

In the physical activity section, the app recommends that the user can perform a minimum of fifteen different physical exercises from a list of possible activities for at least thirty minutes, at least five times per week. The Sehhaty Wa Daghty app provides users with videos and in-depth information regarding all the physical activities included in the app list. In addition to that, at the end of each week, users have the ability to carry out an analysis of the physical activity they have engaged in. This material is intended to assist users in performing the exercises in the correct manner. The app will also send a reminder to the user if they do not accomplish the prescribed level of physical activity or if they do not achieve any physical activity at all (Figure 8).

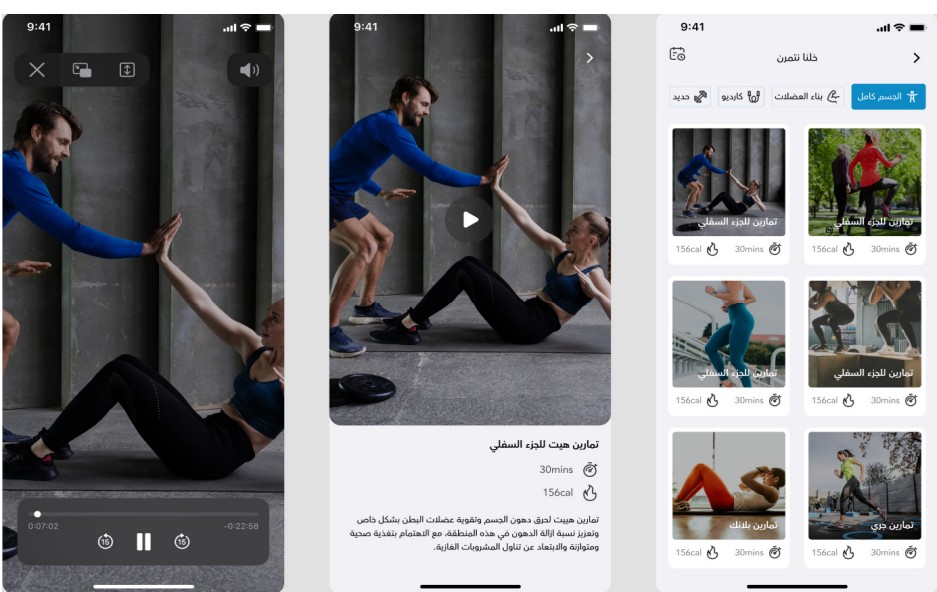

**Figure 8.** Physical activity.

Additionally, the Steps feature is included in the app. It includes a pedometer that enables users to track their daily step count. If the user reaches 5000 steps, the app shows a silver star to the user to encourage the user to walk more. When the user reaches 10,000 steps, a gold star will appear on the screen (Figure 9).

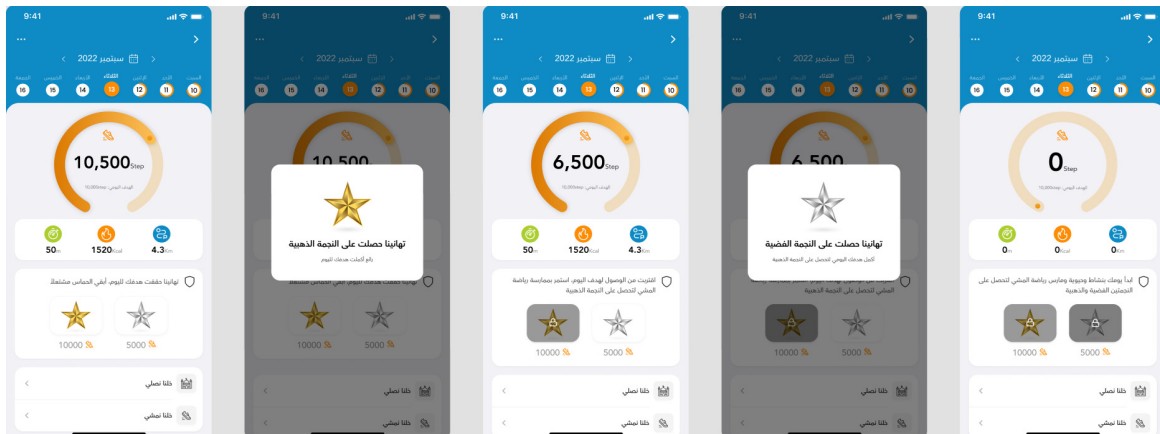

**Figure 9.** Steps.

2.    Related Diet Information

The diet section in Sehhaty Wa Daghty app is divided into four parts: Breakfast, Lunch, Dinner, and Snacks. Therefore, users can organise their daily eating schedules. Most apps available in mobile markets are in English and do not include Saudi meals in their official lists. As a result, Saudi users of these apps will be unable to monitor their calorie intake

because the foods they eat are not included. However, we tried to include many Saudi foods with their calories, such as (Saleeq, Jareesh, Kabsa, etc.). In addition, the user has an opportunity to add any food and enter its calories as well. The number of calories in foods is the indicator by which you can control your weight by keeping it at the required rate without increasing or decreasing it. Therefore, this will make it easy for users to know how many calories are consumed each day. The app offers an interactive interface that allows users to plan or add meals easily (Figure 10).

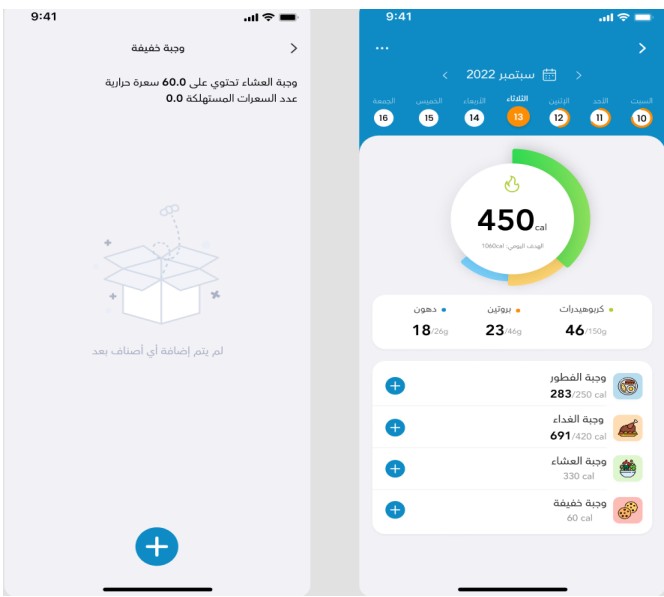

**Figure 10.** Diet template in the app.

In addition, there is a part in the app for water consumption that calculates the amount of water the user should drink daily. If the user drinks four cups of water, equal to one litre, the app shows a silver star to the user, whereas if the user drinks eight cups of water, equal to two litres of water, a gold star will appear on the screen (Figure 11).

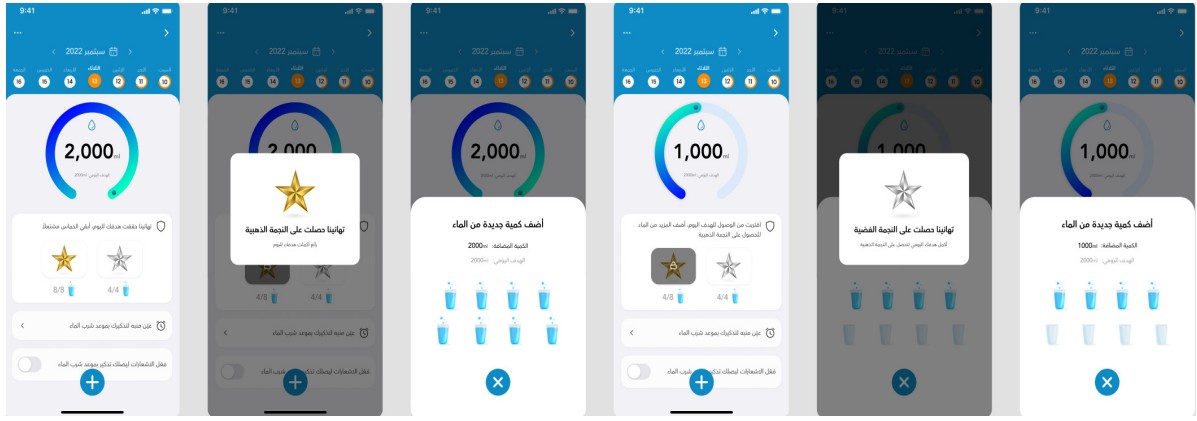

**Figure 11.** Water consumption.

3.    Reminders

The Reminder section in mobile health apps aids in reminding users of the crucial tasks to be completed. The Sehhaty Wa Daghty app sends out reminders to users and assists in keeping track and allows them to choose the date, time, repeat, name, and ringtone for the reminders for daily blood pressure (Figure 12), physical activity (Figure 13), food intake (Figure 14), prayer time (Figure 15), and drinking water (Figure 16).

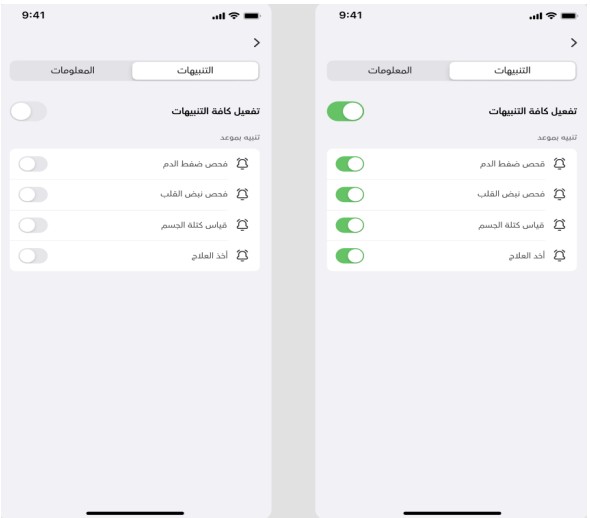

**Figure 12.** Reminders in Sehhaty Wa Daghty app.

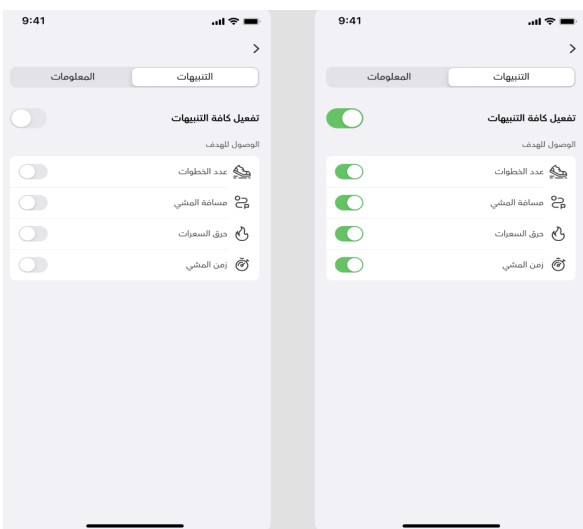

**Figure 13.** Physical activity reminder.

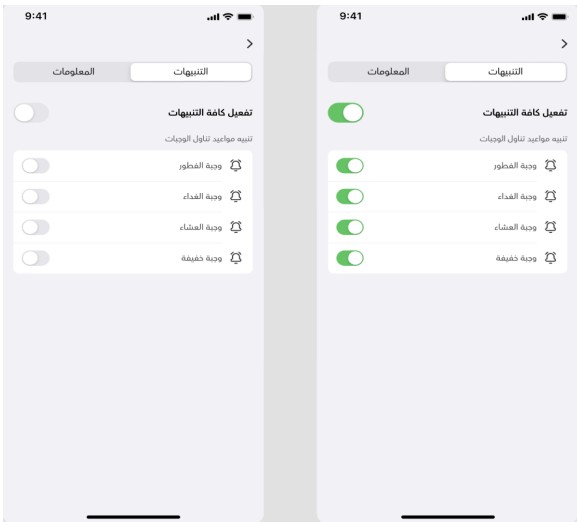

**Figure 14.** Food intake reminder.

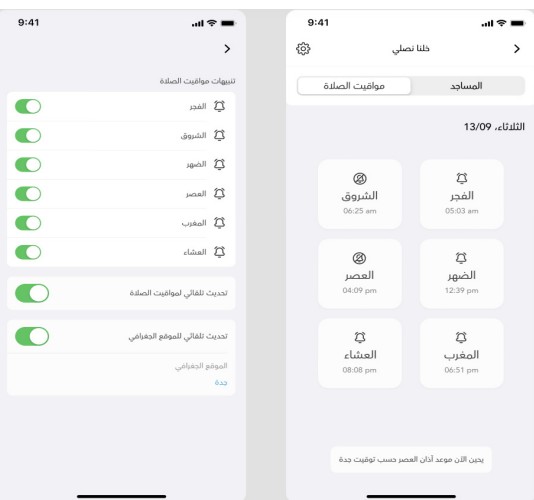

**Figure 15.** Prayer time reminder.

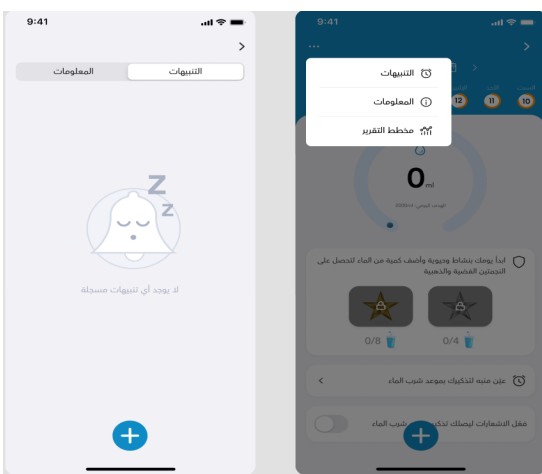

**Figure 16.** Drinking water reminder.

2.2.2. Feature and Usability

This second theme includes the three sub-themes: Sign-Up Process, Social Communication, and Self-Monitoring.

1.    Sign-Up Process

The signing-up process for the Sehhaty Wa Daghty app should be stress-free, mainly because it is the first time the user interacts with the app. Thus, the app should make registration easy, free, and fast. Allowing users to sign up using social network accounts such as Twitter or Facebook can result in an easy or two-click registration procedure (Figure 17). Once the user has an account, they will receive a code sent to their phone number and then can log in quickly to the app (Figure 18).

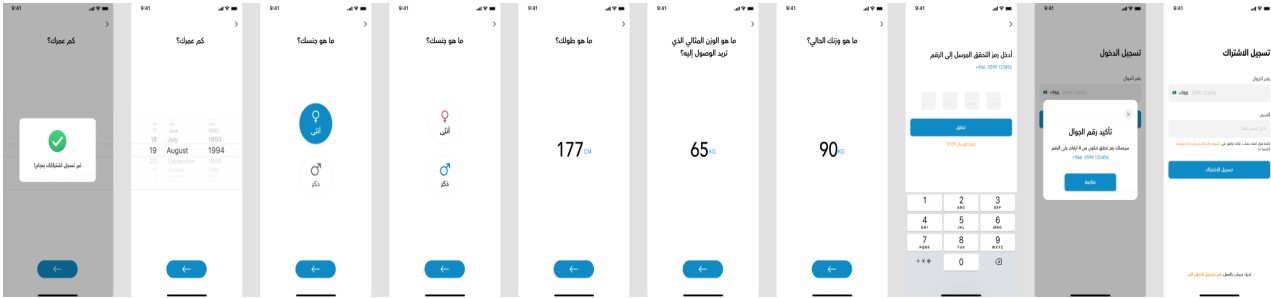

**Figure 17.** Sign-up process.

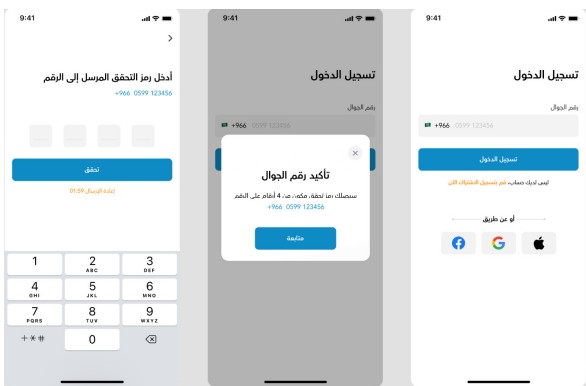

**Figure 18.** Log in to the app.

2. Social Communication

The app allows users to connect to web-based social media platforms (Figure 19). Moreover, it includes a chat feature (Figure 20) so that users may chat with each other about their personal experiences and offer advice to one another, which has a good impact on the users' health behaviour [21]. Users can also get to know one another through this feature, after which they can see each other's progress, speak with one another, post images, and share them across several social media sites such as WhatsApp and Twitter. Real-time communication lets users ask and answer questions in real time [14]. This function will encourage users of the Sehhaty Wa Daghty app to use this app.

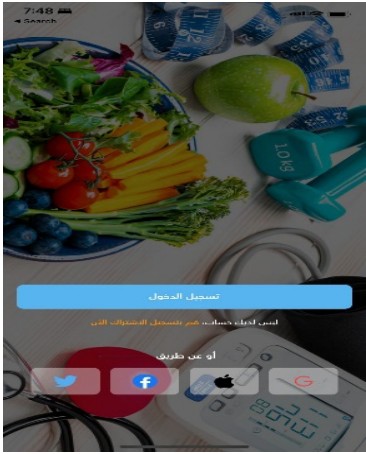

**Figure 19.** Social media access.

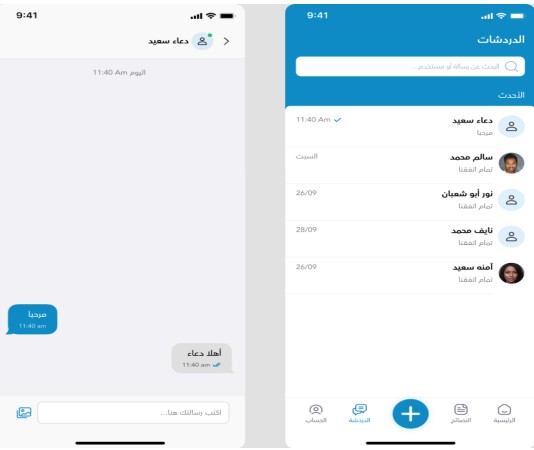

**Figure 20.** Chat between users.

3. Self-Monitoring

Mobile health apps also help individuals set and monitor goals to see changes and keep healthier lifestyles [22]. Therefore, the Sehhaty Wa Daghty app monitors the daily consumption of water, consumption of food (calories), exercise, and step count. At the end of the week, users can conduct a self-assessment of their physical activity, diet, and water intake. In the Sehhaty Wa Daghty app, users can also set goals in many ways, such as distance, time, calories, steps, and so on (Figure 21).

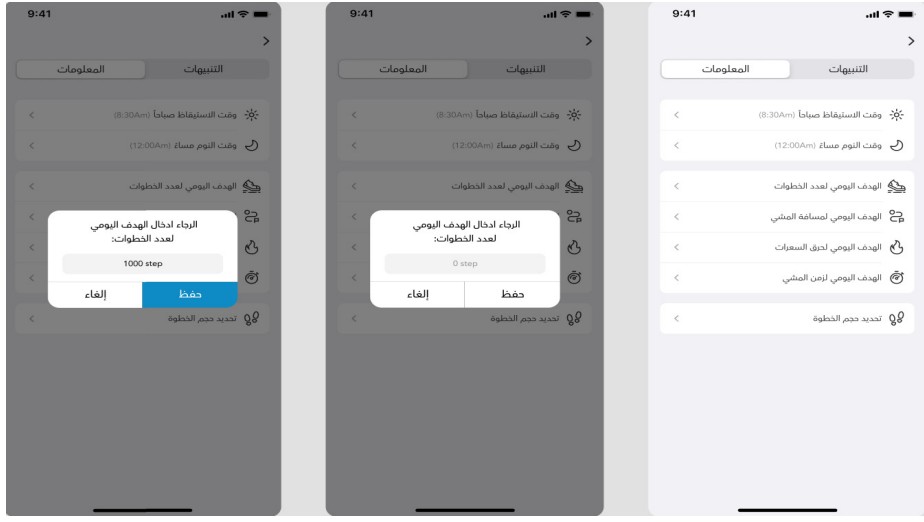

**Figure 21.** Set goals.

2.2.3. Suggestions for App Content

The third theme concentrated on content suggestions for the app. To encourage people to use an application, the application's usability must be considered from the outset and assessed throughout development in order to minimise usability difficulties after the product is launched to the public [23]. The suggestions and expectations of prospective future users should be considered throughout the design and development of a mobile health app for patients with chronic hypertension medical conditions.

Today, more people are using social media and mobile health (mHealth) apps instead of traditional text messaging or email [21]. Family support is another factor that can contribute positively to the self-management of hypertension [24]. Thus, in the Sehhaty Wa Daghty app, patients can see their progress on one page, and the app makes this very easy with charts (Figure 22), the ability to communicate their progress to their family or health care providers (Figure 20), and support for the Arabic language (Figure 23).

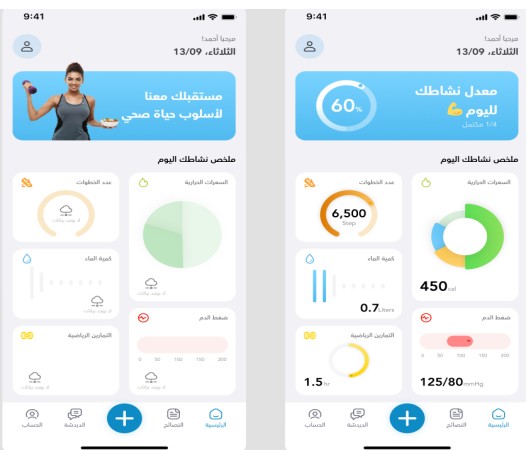

**Figure 22.** App's charts.

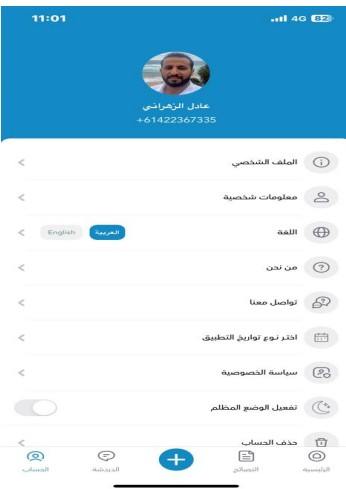

**Figure 23.** Arabic.

The Sehhaty Wa Daghty app provides advice for selecting a workout and planning healthy meals (Figure 24) and has a simple sign-up process with social media (Figure 19).

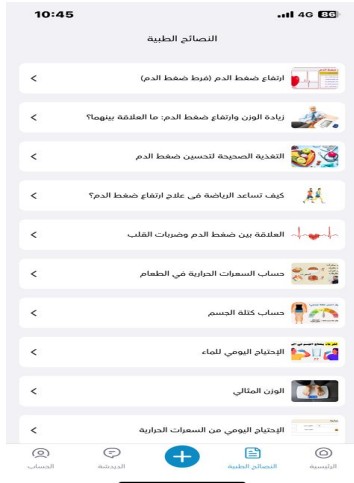

**Figure 24.** Advice section.

The Arabic calendar was also added to the Sehhaty Wa Daghty app as the Saudi community uses it more than the English calendar (Figure 25).

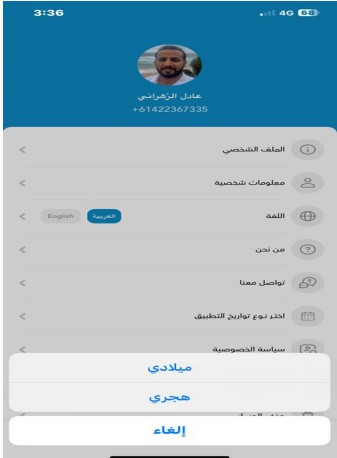

**Figure 25.** Arabic calendar.

2.2.4. Security and Privacy

This is the fourth and last theme in our study. Despite its promise to improve real-time monitoring and healthcare resources, mHealth apps pose data privacy issues due to the sensitive information they can access, sharing user data, and the lack of global privacy regulations [25]. Many users do not use the mobile health application due to the lack of security and privacy of their health information [26,27]. Therefore, in the Sehhaty Wa Daghty app, security and privacy are fundamental aspects when designing the app. The gathering and handling of users' personal health information require specific consideration. By collecting users' personal data, the Sehhaty Wa Daghty app can give users accurate and individualised advice regarding hypertension. Because of the app's nature and usage of private and personal information, the Sehhaty Wa Daghty app follows a number of the principles outlined in the European Commission's Code of Conduct on privacy for mobile health apps [28] to ensure users' security. These principles cover topics such as purpose limitation, data reduction, and user permission.

After signing up for the app, users are asked a series of questions and told how it works before providing their personal information, such as age, gender, and weight. This helps to determine the user's Body Mass Index (BMI), target weight, and diet plan. After creating an account and before using the app, users must consent to the app accessing their health data via the iPhone Health option to gather step, walking, and running data. This information will track users' daily walking distance. The app has a detailed privacy policy that explains why data are collected, what permissions are given, what privacy statements are made, and what information about the app the developers are given. The Sehhaty Wa Daghty app's privacy policy can be found either in the Apple Store or in the app's Settings.

Data encryption should be used to protect user information in the Sehhaty Wa Daghty app. Failure to use data encryption and strong passwords to protect personal health data and secure communication might result in noncompliance with data protection acts, facilitating cybercrime [29]. Encryption in the mobile app is a way to make sure that the data are safe by using algorithms to change plaintext into unreadable text or jumbled code. Ciphertext refers to encrypted data, whereas plaintext refers to unencrypted data [30]. Mobile apps that contain data "at rest" encryption safeguard data on the mobile device (data that are not being transferred) from being accessed by anybody other than the user. Mobile apps that send data to external servers or use cloud storage require a higher level of security (data "in transit" encryption) to prevent unauthorised access [31]. In the Sehhaty Wa Daghty app, a verification code is sent to your phone number once you complete your registration. This protects your personal data and makes sure you are the one who uses the app (Figure 26).

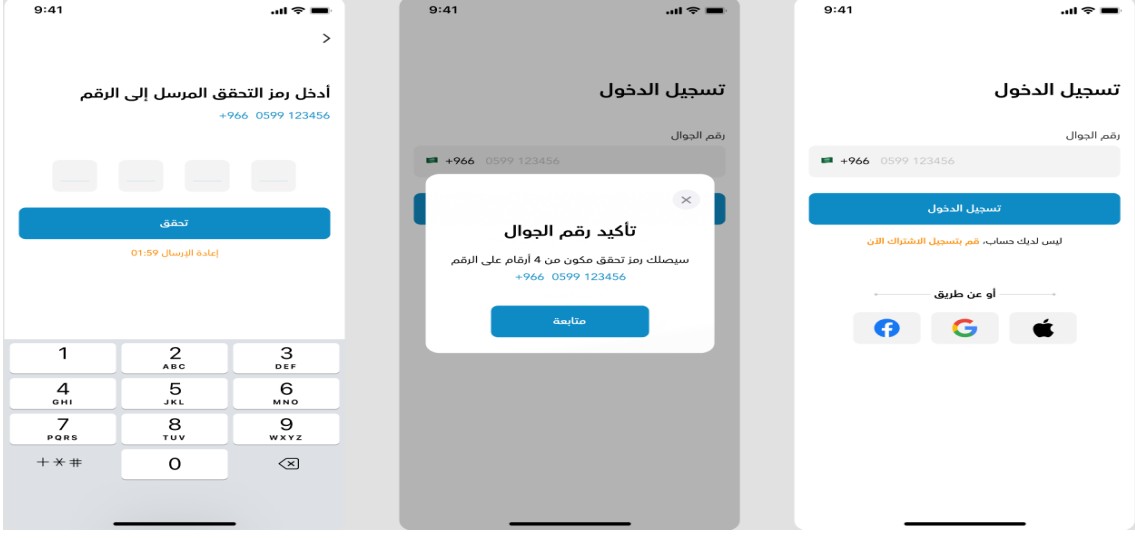

**Figure 26.** Verification code.



## 3. Results

### 3.1. App Testing

While developing the Sehhaty Wa Daghty app and after the initial release, a focus group of five users provided feedback. Table 1 displays their data. All participants are Saudi citizens and suffer from or deal with hypertension. They are very different from one another in terms of age, occupation, and level of education. All the participants were using iPhones. To achieve the research objective, this study reports the emerged features and content needed to be adapted or developed in health apps for hypertension patients during an interactive qualitative analysis focus group activity with ($n$ = 5) experts from the Saudi Ministry of Health. A gap analysis was conducted to develop an app based on a deep understanding of user needs with a patient-centred approach. Moreover, a qualitative semi-structured interview was conducted with stakeholders to obtain information regarding the app in relation to improving the management of hypertension in Saudi Arabia. This audio-recorded interview was held with the focus group to gain their feedback and respond to any queries. The testers were requested to provide feedback about their preferences and suggestions. After two weeks of testing, they gave feedback and suggestions for the app.

The focus group agreed that the Sehhaty Wa Daghty app meets all of the requirements that Saudi hypertensive patients need to self-manage their disease and improve their healthy lifestyles, and that its information is accurate and useful, as explained by one participant:

> "Overall, I like the app, especially the main interface of the app that includes almost everything hypertension patients need such as BP, exercise, steps, water consumption, calories, and percentage of activity today" [P3]

In the Sehhaty Wa Daghty app, you can share your progress with family, friends, and even medical experts in the field of hypertension, as one participant said:

> "Patients can ask questions through chats or share them on social media, so they don't have to go to the doctor for small problems. This makes the app really useful" [P5]

The educational tool inside the Sehhaty Wa Daghty app is an effective tool to help users understand the impact of hypertension in patients' life and how to self-monitor it. One participant indicated that:

> "I like very much the advice section on the app, which is a great idea to know about hypertension disease and the proper way to deal with it without searching in Google but need more information" [P2]

The participants raised a few concerns about the possibility of using voice commands as a form of data entry. One participant acknowledged that:

> " . . . What I liked more about this app that I can navigate via voice command . . . " [P1]

Colour is an essential part of the user-friendly interface. The app will be easier to use and more appealing if the developer chooses the proper colour combination, and the developer should keep in mind that the majority of consumers like straightforward colour schemes that are reflective of their culture. One participant said:

> "The app's colours are wonderful, but what I appreciated more was the dark theme (dark mode)" [P5]

The app should be easy to use and understand, simple, and easy to start using it and learn how to use.

> "The Sehhaty Wa Daghty app is user-friendly and full of helpful information. I like that it presents my health data in understandable charts" [P1]

Physical activity is thought to be an effective strategy to improve behaviour in order to self-manage hypertension. Most participants were pleased with the app's physical activities

feature, which allows them to log their daily steps in order to manage their blood pressure and enhance the self-monitoring of hypertension. They prefer to view videos before exercising and require additional activity to be added to the app. Two participants stated:

> "I would like to add more exercise in the app, such as swimming, as I have a private swimming pool in my house and as well as cycling" [P4]

> "I enjoy how each activity in the app has videos and detailed information on all of the physical activities featured in the app list" [P2]

### 3.2. Suggestions for the App

The potential users and medical experts mentioned many points to improve the Sehhaty Wa Daghty app. Experts and some users suggest adding AI to the app as some users need to use voice technology in the app (Figure 27). Voice technology was introduced to iPhone users in 2011 via Siri. As with the most prominent IT services giants, Google and Amazon have introduced AI-enabled virtual assistants to compete in the marketplace [32]. Voice assistants have developed over time, becoming more efficient at meeting the demands of their users. A voice assistant is an AI-enabled virtual assistant that detects the user's voice and employs natural language processing algorithms for speech synthesis to carry on complicated real-time communications with the user [33]. These are some voice commands that have been included in the Sehhaty Wa Daghty app till now:

1. Add medicine;
2. Add BP;
3. Add water;
4. Add meal;
5. Water consumed;
6. Water remaining;
7. Calories consumed;
8. Calories remaining;
9. Steps count;
10. Chart BP.

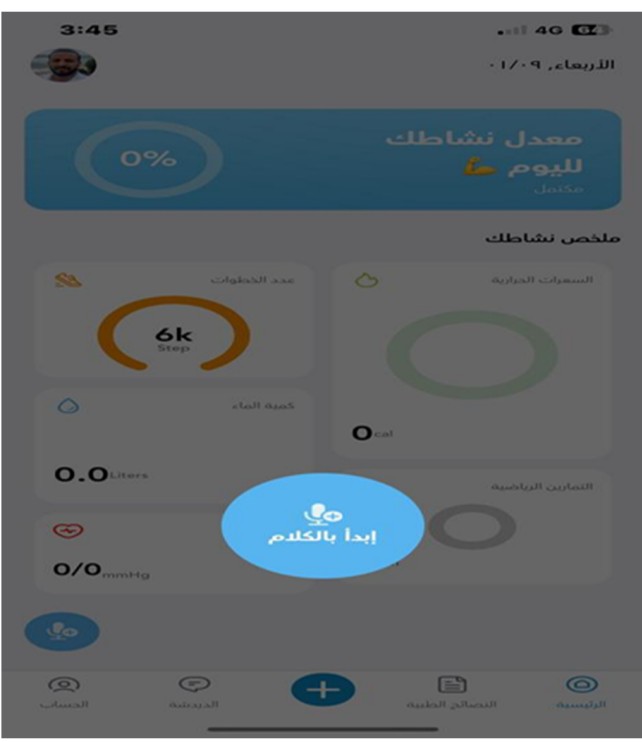

**Figure 27.** AI in the Sehhaty Wa Daghty app.

## 4. Discussion

### 4.1. Principal Findings

The Sehhaty Wa Daghty application is an interactive, user-friendly app specially built for iPhones. It includes several functions designed to help users monitor and manage their blood pressure, food consumption, and physical activities. The app offers individualised diet and hypertension control management guidance. Let's Walk, Let's Pray, and Let's Train are unique features designed to encourage users to walk and exercise more. The application also uses Artificial Intelligence (AI) to enable users to engage with the apps and obtain the information they require, fast and conveniently. The Sehhaty Wa Daghty app takes user privacy and data security seriously by following a set of standards and procedures.

Furthermore, this study aims to explain the steps that were taken to develop a hypertension app (the Sehhaty Wa Daghty app) that includes motivational features, Saudi male and female needs, interactive design, and social and cultural values [4]. The Sehhaty Wa Daghty app has some unique features such as the following:

1. Medical experts and end users collaborated on the design of the Sehhaty Wa Daghty app, and their requirements, feedback, and recommendations were incorporated into the app's beta version;
2. The app provides access to social networks so that users can communicate with one another;
3. The app's design considers several aspects of usability, especially the easy-to-use attribute;
4. The application includes calorie counts for a variety of Saudi dishes;
5. Through a specialised database, users have the ability to view the history of their calorie consumption as well as the amount of effort they have put in;
6. The application is compatible with voice commands using AI;
7. Using the Let's Walk function, the app motivates users to participate in group walks;
8. Using the Let's Pray function, users can be directed to the closest mosques;
9. Using the Let's Train function, users are encouraged to train regularly;
10. The app supports the Arabic calendar, which is the official calendar in Saudi Arabia.

### 4.2. Potential Impact

Hypertension rates are increasing in Saudi Arabia because people do not know enough about the disease, or suitable nutrition, do not exercise enough, and cannot perform many physical activities outside because of the hot weather. The study, supported the importance of promoting physical activity as a key issue in preventing non-communicable diseases, including hypertension [34]. Indeed, the study pointed to sedentary living and physical inactivity as critical issues of chronic illnesses, including hypertension prevention in Saudi Arabia. A typical response was that, while there are some existing initiatives promoting physical activity, there is a need for sustained health promotion programmes for increasing physical activity. Moreover, another study [35] highlighted the need for a greater focus on policies and the promotion of physical activity among adults, including greater attention to ensuring environmental changes that facilitated more physical activity. Other research in Saudi Arabia also identified the need for more effective programmes to promote healthy nutrition and physical activity, including better access to dietitians/nutritionists [36]. Accordingly, smartphones and their applications have become popular in Saudi Arabia, and this is a challenge for developers to design a suitable app, especially for patients. Mobile health applications have resulted in various positive effects for patients [1]. The Sehhaty Wa Daghty app aids Saudi adults with hypertension in self-managing their condition and enhancing their health and fitness levels. Based on the results of the potential users' and medical experts' reviews and feedback, it is predicted that using the Sehhaty Wa Daghty app will help patients with hypertension manage their disease in Saudi Arabia.

First, the app is free, because the number of downloads for free apps is more than 10 times that of paid apps [37,38]. Second, motivational features such as the Let's Pray feature will encourage users to walk while they go to the mosque, as 93% of the country

believes in Islam. Muslims have to pray five times daily [39], so this feature encourages users to walk to the mosque instead of using their cars. The Let's Walk feature will gather users to walk in a group instead of walking individually. This will result in significant benefits for users, reminding them to walk in order to reduce their weight and burn more calories. According to [40], walking in nature with a group on a regular basis was connected with an increase in positive affect and a better overall sense of health and well-being, as well as a reduction in stress levels, depression, and negative affect. The Let's Train feature helps users to train regularly with aerobic exercises that are recommended by medical experts for hypertension patients to reduce their high blood pressure and to self-manage the disease [41,42].

The app also includes a pedometer that lets users keep track of the number of steps they take each day. When a user reaches their daily step target, a star of either silver or gold shows on the screen to acknowledge their achievement. It will be possible for users to discuss their accomplishments with their colleagues or family members using the built-in chat feature or using social media platforms. This will have a beneficial effect on the users' behaviour towards their health. Furthermore, including Saudi native food varieties, offering their calories, and the option to easily add a new food item to the app are additional features that can encourage app users to eat healthy food and monitor their diets. These features, as well as the app's ease of use, will positively contribute to the target group's acceptance of the app and assist in managing and controlling hypertension in Saudi Arabia.

*4.3. Strengths*

The Sehhaty Wa Daghty app identifies the feedback and recommendations of the initial focus group, which enables the app to meet the specific requirements of hypertensive Saudi people through a localised and tailored method based on the perspectives and needs of Saudi individuals regarding the design of a hypertension management mobile app to be used by hypertension patients to better manage their illnesses in our previous paper [4]. To the authors' knowledge, this is the first app that provides a new Arabic hypertension app that is created with usability characteristics, motivating features, and social and cultural factors with the assistance of medical specialists and end users. This application will eventually help Saudi patients to control and self-assess this disease.

*4.4. Limitations*

Rather than generalize findings, this qualitative research intended to understand focus group participants and case study interviews. This study did not compare the findings to other mHealth apps; this study was limited to the expert in Jeddah City who participated in the focus group and case study. Moreover, other limitations of the Sehhaty Wa Daghty app include the following: It is not available on phones other than the iPhone, such as Android and BlackBerry. The app also does not work with virtual reality features or smartwatches such as Apple Watches or fitness trackers such as Fitbits at the moment. It does not let you scan food barcodes, and the app does not work when you're not connected to the Internet. Chatbots, which are healthcare virtual assistant services, do not exist in the Sehhaty Wa Daghty app, which can be used to provide basic healthcare services and can also help patients in many ways, such as setting up appointments, answering common questions, helping with the payment process, and even giving basic virtual diagnostics [18]. However, in the near future, finding ways to address these problems and regularly updating the app will help to keep users interested and motivated.

*4.5. Next Steps*

A quantitative study will be undertaken with a group of 10 Saudi citizens from the city of Jeddah to better identify the Sehhaty Wa Daghty app's usability level and improve it. They will test the usability using the PACMAD Usability model chosen for this application [43]. The usability attributes in the PACMAD model that will be tested are

effectiveness, efficiency, satisfaction, learnability, memorability, errors, cognitive load, and the user experience (UX), for designing the Sehhaty Wa Daghty app. In addition, creating the software for other major mobile phone platforms, such as Android, and integrating the app with smartwatches are future goals.

**Author Contributions:** Supervision, V.G. and R.A.; writing—original draft, A.A.; writing—review & editing, V.G. and R.A. All authors have read and agreed to the published version of the manuscript.

**Funding:** This research received no external funding.

**Data Availability Statement:** The data presented in this study are available upon request from the corresponding author. The data are not publicly available due to privacy restrictions.

**Conflicts of Interest:** The authors declare no conflict of interest.

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
