# Peer review of "Enabled Artificial Intelligence (AI) to Develop Sehhaty Wa Daghty App of Self-Management for Saudi Patients with Hypertension: A Qualitative Study"

_information, doi:10.3390/info14060334_

Round 1

Reviewer 1 Report

The article presents the process of developing an app for smartphones (IOS only) for use in cases of hypertension for people in Saudi Arabia. The app includes BP monitoring, nutrition, and physical activity tracking. It only uses information coming from the smartphone without connecting to other devices. The app was developed with the involvement of doctors and patients and support from the Saudi Arabian Ministry of Health. Allows access to social networks. It includes specifics such as Saudi Arabian dishes. It was designed for a Muslim population. Uses verbal commands and AI. The article describes the two phases of the app's development, the first using interviews with 21 participants and the second, a focus group with 5 specialists, aimed at identifying implementation instruments and development resources.

The App receives manually entered information from Arterial Pressure, amount of water, food, and steps. Provides tracking and reminders related to treatment, physical activity, and water intake. It also provides indications of a religious nature

The text is well written and is quite detailed, the references are up to date, the topic is current and although there is development in this matter, the development of the specifics is interesting.

Developing an app specifically for specific populations makes sense and seems to be a strategy to follow. Engaging users, clinicians, and patients include their perspectives and allows for better matching. Using an inclusive perspective of the main components of the therapeutic approach to hypertension is suitable for the holistic approach to hypertension.

There are no supporting references to the Physical Activity block, only a reference to guidelines from the Ministry of Health,  scientific support would be convenient.

It is not clear how the information is shared with the care team. The self-care component is relevant, as well as the concern with usability.

In some frameworks, this application is considered a medical device; this aspect is not developed.

Reflection on potential impact, strengths, limitations, and next steps is realistic

In addition to the convenience of presenting the referential support of the physical activity component

I also suggest that line 41 references “According to [2]” should be written otherwise (the sentence with the reference number), as happens in most of the other references. There are others with the same problem (ln 46)

The word “preference” (ln 45) is probably “prevalence.”

Author Response

We greatly appreciate the reviewers’ enthusiasm about manuscript and the quality comments offered by the reviewers. The suggestions offered by the reviewers have been immensely helpful, and we also appreciate your insightful comments on revising the paper. We have included the reviewers’ comments in the following table and responded to them individually, describing the changes we have made. Also, all changes are highlighted and have been incorporated to the manuscript, and the revised manuscript is attached.

Author's Reply to the Review Report (Reviewer 1)

The article presents the process of developing an app for smartphones (IOS only) for use in cases of hypertension for people in Saudi Arabia. The app includes BP monitoring, nutrition, and physical activity tracking. It only uses information coming from the smartphone without connecting to other devices. The app was developed with the involvement of doctors and patients and support from the Saudi Arabian Ministry of Health. Allows access to social networks. It includes specifics such as Saudi Arabian dishes. It was designed for a Muslim population. Uses verbal commands and AI. The article describes the two phases of the app's development, the first using interviews with 21 participants and the second, a focus group with 5 specialists, aimed at identifying implementation instruments and development resources.

The App receives manually entered information from Arterial Pressure, amount of water, food, and steps. Provides tracking and reminders related to treatment, physical activity, and water intake. It also provides indications of a religious nature.

The text is well written and is quite detailed, the references are up to date, the topic is current and although there is development in this matter, the development of the specifics is interesting.

Thanks for your supportive comments…

Developing an app specifically for specific populations makes sense and seems to be a strategy to follow. Engaging users, clinicians, and patients include their perspectives and allows for better matching. Using an inclusive perspective of the main components of the therapeutic approach to hypertension is suitable for the holistic approach to hypertension.

Thanks for your supportive comments…

There are no supporting references to the Physical Activity block, only a reference to guidelines from the Ministry of Health,  scientific support would be convenient.

Thanks for your comments and some references have been added accordingly and incorporated into the manuscript.

In their study, [34] supported the importance of promoting physical activity as a key issue for preventing non-communicable diseases, including hypertension. Indeed, the study pointed to sedentary living and physical inactivity as critical issues of chronic illnesses, including hypertension prevention in Saudi Arabia. A typical response was that while there are some existing initiatives promoting physical activity, there is a need for sustained health promotion programmes for increasing physical activity. Also, another study [35] highlighted the need for a greater focus on policies and the promotion of physical activity among adults, including greater attention to ensuring environmental changes that facilitated more physical activity. Other research in Saudi Arabia also identified the need for more effective programs to promote healthy nutrition and physical activity, including better access to dietitians/ nutritionists [36]. Accordingly, smartphones and their applications have become popular in Saudi Arabia, and this is a challenge for developers to design a suitable app, especially for patients. Mobile health applications have resulted in various positive effects for patients [1]. The Sehhaty Wa Daghty app aids Saudi adults with hypertension in self-managing their condition and enhancing their health and fitness levels. Based on the results of the potential users' and medical experts' reviews and feedback, it is predicted that using the Sehhaty Wa Daghty app will help patients with hypertension manage their disease in Saudi Arabia.

It is not clear how the information is shared with the care team. The self-care component is relevant, as well as the concern with usability.

Thanks for your comments; this has been noted and incorporated in the manuscript. The information is exchanged between healthcare team members in a structured, concise, and accurate manner to ensure safe patient care.

In addition, the second phase of this study will consist of testing the app's usability, which will include testing for the following seven attributes of PACMAD model: effectiveness, efficiency, satisfaction, cognitive load, learnability, errors, and memorability. 

In some frameworks, this application is considered a medical device; this aspect is not developed.

At our research stage, our mHealth app is a mobile application designed to promote wellness and manage chronic diseases, promote healthy behaviours, quality of life, and well-being of hypertension patients’ reflection on potential impact. It is not a medical device as it is not used to diagnose and treat hypertension, but it assists in providing decision-support and collecting health-related information.

Reflection on potential impact, strengths, limitations, and next steps is realistic.

Thanks for your supportive comments…

In addition to the convenience of presenting the referential support of the physical activity component

Thanks for your supportive comments…

I also suggest that line 41 references “According to [2]” should be written otherwise (the sentence with the reference number), as happens in most of the other references. There are others with the same problem (ln 46)

Thanks for your comments. These comments have been modified accordingly.

The word “preference” (ln 45) is probably “prevalence.”

Thanks for your comment. Yes, you are right it is prevalence, and this has been amended accordingly.

Reviewer 2 Report

This paper presents a mHealth system aimed at helping the management of weight and well-being in hypertensive patients in Saudi Arabia. The system is thoroughly presented, all modules and functionalities described, and the possibilities for interventions from the attending physicians, as well as closed loop coaching and inter-patient interactions are presented in detail. Although the system presented has some interesting features (e.g. helping patients socialise and perform group activities) the authors should address some issues before resubmitting.

1) One problem is the small number of patients who used and assessed the system. The user acceptance as well as the measures for evaluating the system performance need considerable effort, large number of users and the use of standard useability tools (e.g. SUS, neasurements of hedonic deficits, etc). How do the authors plan to address in the future these issues?

2) In my view one of the critical downfalls of such systems is the issue of nutrition that is addressed here. The system is not based on objective measurements of food intake or other measureable features such as rate of eating , type of food consumed (e.g. ultra-processed food etc). I would urge the authors to address these gaps in the discussion and perhaps get input from a couple of EU projects in the area such as SPLENDID and BigO (Maramis et al, CMPB, 2020, Dhammawati et al, Nutrients, 2023, Tragomalou et al, Nutrients, Fagerberg et al, Nutrients, 2021). and/or other analogous projects with more objective measurements aiding in the assessment of the patients' condition via telemonitoring and eHealth applications.

3) Finally do the patients measure their BP and enter it in the system as shown once a day, or is there the plan to use wearables to continuously measure BP in the hypertention patients along with heart rate and combine HRV and BP analytics with the behavioral factors acquired from the presented mHealth app?

English is OK.

Author Response

We greatly appreciate the reviewers’ enthusiasm about the manuscript and the quality comments offered by the reviewers. The suggestions offered by the reviewers have been immensely helpful, and we also appreciate your insightful comments on revising the paper. We have included the reviewers’ comments in the following table and responded to them individually, describing the changes we have made. Also, all changes are highlighted and have been incorporated into the manuscript, and the revised manuscript is attached.

Author's Reply to the Review Report (Reviewer 2)

This paper presents a mHealth system aimed at helping the management of weight and well-being in hypertensive patients in Saudi Arabia. The system is thoroughly presented, all modules and functionalities described, and the possibilities for interventions from the attending physicians, as well as closed loop coaching and inter-patient interactions are presented in detail. Although the system presented has some interesting features (e.g. helping patients socialise and perform group activities) the authors should address some issues before resubmitting.

  • One problem is the small number of patients who used and assessed the system. The user acceptance as well as the measures for evaluating the system performance need considerable effort, large number of users and the use of standard useability tools (e.g. SUS, neasurements of hedonic deficits, etc). How do the authors plan to address in the future these issues?

Thanks for your comments! The small number is just a sample. It is a focus group of 5 expert participants. It was conducted to collect current and emerging provider-focused implementation tools and resources for developing an app that meets the needs and preferences of hypertension patients. I have ethical approval for it. This paper is apart from on progress research project which focuses on exploring individuals’ perspectives and needs to design hypertensive management technology solution.

The second step of this study will consist of testing the app's usability, which will include testing for the following seven attributes of PACMAD model: effectiveness, efficiency, satisfaction, cognitive load, learnability, errors, and memorability. The usability testing outcomes will show how well the app performs and what the target audience needs and expects. This is essential for such an application's success because the input is gathered from the end users.

  • In my view one of the critical downfalls of such systems is the issue of nutrition that is addressed here. The system is not based on objective measurements of food intake or other measureable features such as rate of eating , type of food consumed (e.g. ultra-processed food etc). I would urge the authors to address these gaps in the discussion and perhaps get input from a couple of EU projects in the area such as SPLENDID and BigO (Maramis et al, CMPB, 2020, Dhammawati et al, Nutrients, 2023, Tragomalou et al, Nutrients, Fagerberg et al, Nutrients, 2021). and/or other analogous projects with more objective measurements aiding in the assessment of the patients' condition via telemonitoring and eHealth applications.

Thanks for your comments! The nutrition policies have been highlighted, included in the discussion, and supported with references. The app is designed for Saudi hypertension patients, and the food is the traditional food in Saudi Arabia and was approved by the experts at the Ministry of Health. The nutrition policies have been articulated in the discussion as follows. A range of policies led by the MOH and Saudi Food and Drug Authority (SFDA) were identified that are relevant to the healthy food strategy to eliminate nutrition-related risk factors for chronic disease patients, including hypertension patients. These policies are as follows:  limiting Trans Fatty Acids (TFA) to 2% of the composition for fats and oils and 5% for other products, mandatory TFA labelling in the food supply, which came into effect in 2015, and a ban on partially hydrogenated oils (PHOs) enacted in 2018; sodium limit of 1/100 g for bakery products. These policies reflect the objective of the Saudi Arabian government to comply with and align legislation with the WHO global action plan for the prevention and control of chronic diseases, including hypertension.

  • Finally do the patients measure their BP and enter it in the system as shown once a day, or is there the plan to use wearables to continuously measure BP in the hypertention patients along with heart rate and combine HRV and BP analytics with the behavioral factors acquiredfrom the presented mHealth app?

Thanks for your comments! The main focus right now is that patients measure their PB daily and enter it into the app manually, and in the future, we plan to measure BP and heart rate via wearable devices connected to the app. Although we are aware that wearable devices will make the app easy to continuously measure BP in hypertension patients along with heart rate, not all users have access to wearables. In addition, creating the software for other major mobile phone platforms, such as Android, and integrating the app with smartwatches are future goals as you can see in next steps section.

Reviewer 3 Report

Thank you for the opportunity to review this manuscript by Adel Alzahrani and colleagues.

The aim of the paper is to present the methodology involved in developing a hypertension app that takes into account: cultural and social standards in Saudi Arabia, the impact of motivational features, and the needs of male and female Saudi citizens.

The topic is interesting and relevant. The structure of the paper has a clear and logical flow. Strengths of the Sehhaty Wa Daghty app are identified, as well as limitations leading to next steps.

I have some comments/suggestions in order to improve the quality of this paper:

1.      The authors did not respect the request stated in the Microsoft Word Template for Information Journal papers : “Abstract: A single paragraph of about 200 words maximum” (see https://www.mdpi.com/journal/information/instructions, “Accepted File Formats“ chapter);

2.      The originality/novelty of the approach should be emphasised at the end of the Introduction. In particular, it would be useful if the authors explained what this paper adds to the subject area compared with other published materials on the same topic. This request is (partially) fulfilled in lines 556-558 of the paper;

3.      In relation to the open-ended  interview qualitative questionnaire used for the 21 participants in the first phase, please specify:

·         How the informants were recruited;

·         Were there differences in the results based on the age or other characteristics of the informants?

·         Whether respondents had a guide for the future interviews;

·         Whether written records or transcripts of audio records have been returned to all the participants for comments and/or corrections;

·         Whether respondents have been invited to provide feedback on the interview process;

·         Whether the open-ended interview questions to guide the discussion had been approved by authorised persons;

4.      Table 1 does not respect the format requested in the Microsoft Word Template for Information Journal papers (see https://www.mdpi.com/journal/information/instructions, “Accepted File Formats“ chapter);

5.      The clarity/accuracy of the figures should be improved;

6.      There are some problems in using abbreviations:

·         The abbreviation “smartphone applications (apps)” is defined in line 59 and used for the first time in line 19, before its definition;

·         The abbreviation “Artificial intelligence (AI)” is defined in line 70 and redefined in line 71-72;

·         The abbreviation BMI (line 380) is not defined;

7.      On lines 77-78, in the phrase “We included AI in the [ Sehhaty Wa Daghty] app, which mean”, the authors have to make the predicate-subject agreement;

8.      In Line 520, in the text “[1]. he Sehhaty Wa Daghty app aids” – a “T” is missing;

9.      In lines 556, the text “To the author's knowledge” should be changed to “To the authors’ knowledge” because the paper has 3 authors;

10.  The last chapter of the paper “7. Next step” should be named “7. Next steps” because there are 2 steps included in it;

11.  The description of the references should respect the requirements specified in the instructions for the authors of Information journal (https://www.mdpi.com/journal/information/instructions);

For these reasons, I would recommend a major revision of the manuscript.

I hope my feedback is useful to the authors in improving their paper and wish them all the best in pursuing this important area of research.

Author Response

We greatly appreciate the reviewers’ enthusiasm about the manuscript and the quality comments offered by the reviewers. The suggestions offered by the reviewers have been immensely helpful, and we also appreciate your insightful comments on revising the paper. We have included the reviewers’ comments in the following table and responded to them individually, describing the changes we have made. Also, all changes are highlighted and have been incorporated into the manuscript, and the revised manuscript is attached.

Author's Reply to the Review Report (Reviewer 3)

Thank you for the opportunity to review this manuscript by Adel Alzahrani and colleagues.

The aim of the paper is to present the methodology involved in developing a hypertension app that takes into account: cultural and social standards in Saudi Arabia, the impact of motivational features, and the needs of male and female Saudi citizens.

The topic is interesting and relevant. The structure of the paper has a clear and logical flow. Strengths of the Sehhaty Wa Daghty app are identified, as well as limitations leading to next steps.

I have some comments/suggestions in order to improve the quality of this paper:

  1. The authors did not respect the request stated in the Microsoft Word Template for Information Journal papers : “Abstract: A single paragraph of about 200 words maximum” (see https://www.mdpi.com/journal/information/instructions, “Accepted File Formats“ chapter);

Thanks for your valuable comments, these comments have been addressed accordingly, please see the Abstract:

The prevalence of uncontrolled hypertension is rising all across the world, making it a concern for public health. The usage of mobile health applications has resulted in a number of positive outcomes for management and control of hypertension.

The study's primary goal is to explain the steps to create a hypertension app that considers cultural and social standards in Saudi Arabia, motivational features and the needs of male and female Saudi citizens.

This study reports the emerged features and content needed to be adapted or developed in health apps for hypertension patients during an interactive qualitative analysis focus group activity with (n=5) experts from the Saudi Ministry of Health. A gap analysis was conducted to develop an app based on deep understanding of user needs with patient-centred approach.

Based on the participant's reviews in this study, the app was easy to use, can help Saudi patients to control their hypertension, the design was interactive, motivational features are user-friendly and need to consider other platforms such as Android and blackberry in a future version.

Mobile health apps can help Saudis change their unhealthy lifestyles. Target users, usability, motivational features, and social and cultural standards must be considered to meet the app's aim.

  1. The originality/novelty of the approach should be emphasised at the end of the Introduction. In particular, it would be useful if the authors explained what this paper adds to the subject area compared with other published materials on the same topic. This request is (partially) fulfilled in lines 556-558 of the paper;

Thanks for your valuable comments, these comments have been addressed accordingly.

This paper aims to explain the development of the Sehhaty Wa Daghty app of self-management for patients with hypertension in Saudi Arabia and how it can use AI to help patients to manage their diseases. The Sehhaty Wa Daghty app aids Saudi adults with hypertension in self-manage their condition and enhancing their health and fitness levels.

  1. In relation to the open-ended  interview qualitative questionnaire used for the 21 participants in the first phase, please specify:
  • How the informants were recruited;

A total of 21 participants were identified through the researchers’ contacts and were interviewed. A snowballing technique was employed where initial respondents identified other relevant colleagues who were referred to the study. Participants were eligible to participate if they had worked or had experience in hypertension management. Additionally, the inclusion criteria included patients who had had hypertension disease for at least one year and were older than 18 years. All participants meeting these criteria were approached to participate in the study. All interviewees were contacted through email, with study information provided.

  • Were there differences in the results based on the age or other characteristics of the informants?

There are some characteristics agreed upon in general for all of them but in detail that elderly users requested, such as easy to use, the addition of a dark mode and voice assistance.

  • Whether respondents had a guide for the future interviews;

After reviewing relevant literature on chronic diseases and the healthcare system in Saudi Arabia, a semi-structured interview guide was developed to explore the participants’ views about the demands and requirements of the users, as well as the usability rules, norms, and culture of mHealth apps. The interview type was semi-structured to allow the interview to be responsive to respondents’ answers and concepts emerging throughout the data collection process. All interviewees were contacted through email, with study information provided.

  • Whether written records or transcripts of audio records have been returned to all the participants for comments and/or corrections;

Participant validation was achieved by returning transcripts to interviewees after the completion of the interviews for comments or corrections.

  • Whether respondents have been invited to provide feedback on the interview process;

Researchers asked participants for feedback during interviews to improve study quality and efficacy. This feedback can reveal interviewees' viewpoints, pleasures, and concerns. Also, the interviewer ensured that the respondents' feedback was considered and that they fully understood the questions being asked and responded in the appropriate context with appropriate feedback.

  • Whether the open-ended interview questions to guide the discussion had been approved by authorised persons;

Informed written consent was taken from all participants prior to interviews. Also, the ethics committee of the University of Technology Sydney, UTS HREC REF NO granted the approval to conduct this study. ETH21-6571.

  1. Table 1 does not respect the format requested in the Microsoft Word Template for Information Journal papers (see https://www.mdpi.com/journal/information/instructions, “Accepted File Formats“ chapter);

Thanks for your valuable comment. This comment has been addressed accordingly. Please see Table 1

  1. The clarity/accuracy of the figures should be improved;

Thanks for your valuable comment. This comment has been addressed accordingly.

  1. There are some problems in using abbreviations:
  • The abbreviation “smartphone applications (apps)” is defined in line 59 and used for the first time in line 19, before its definition;

Thanks for your valuable comment. This comment has been addressed accordingly, please see line 19

  • The abbreviation “Artificial intelligence (AI)” is defined in line 70 and redefined in line 71-72;

Thanks for your valuable comment. This comment has been addressed accordingly, please see line 71-72

  • The abbreviation BMI (line 380) is not defined;

Thanks for your valuable comment. This comment has been addressed accordingly, please see line 380

  1. On lines 77-78, in the phrase “We included AI in the [ Sehhaty Wa Daghty] app, which mean”, the authors have to make the predicate-subject agreement;

Thanks for your valuable comment. The phrase has been addressed accordingly as “We included AI in the [ Sehhaty Wa Daghty] app, which means” please see line 77-78

  1. In Line 520, in the text “[1]. he Sehhaty Wa Daghty app aids” – a “T” is missing;

Thanks for your valuable comment. This comment has been addressed accordingly, please see line 520

  1. In lines 556, the text “To the author's knowledge” should be changed to “To the authors’ knowledge” because the paper has 3 authors;

Thanks for your valuable comment. This comment has been addressed accordingly, please see line 556

  1. The last chapter of the paper “7. Next step” should be named “7. Next steps” because there are 2 steps included in it;

Thanks for your valuable comment, this comment has been addressed and incorporated in the manuscripts under the Next steps subheading. Please see the green highlight in the manuscript.

  1. The description of the references should respect the requirements specified in the instructions for the authors of Information journal (https://www.mdpi.com/journal/information/instructions);

The references section has been revised accordingly.

Round 2

Reviewer 2 Report

The authors addressed the revisions appropriately.

English is fine.